# The Health Impacts of Hazardous Chemical Exposures among Child Labourers in Low- and Middle-Income Countries

**DOI:** 10.3390/ijerph18105496

**Published:** 2021-05-20

**Authors:** Natasha B. Scott, Nicola S. Pocock

**Affiliations:** 1LSHTM Alumni, London SW1V 4LS, UK; 2Lumos Foundation, London EC3R 8NB, UK; nicola.pocock@wearelumos.org; 3Gender Violence & Health Centre, London School of Hygiene and Tropical Medicine, London WC1E 7HT, UK

**Keywords:** child labour, hazardous work, chemical exposure, health impact

## Abstract

Of 218 million working children worldwide, many are suspected to be exposed to hazardous chemicals. This review aims to synthesize reported evidence over the last two decades on chemical exposure and adverse health consequences in children labourers in low- and middle-income Countries (LMIC). Included studies investigated health outcomes related to chemical exposures among child labourers aged 5–18 in LMIC. Twenty-three papers were selected for review, focusing on pesticides (*n* = 5), solvents (*n* = 3), metals (*n* = 13) and persistent organic pollutants (POPs) (*n* = 2). Adverse health effects identified among child labourers included abnormal biomarkers, for example elevated blood and urine chemical concentrations, neurobehavioural deficits and neurological symptoms, mental health issues, oxidative stress and DNA damage, poor growth, asthma, and hypothyroidism. Workplace exposure to chemicals has pernicious health effects on child labourers. Large research gaps exist, in particular for long-term health impacts through chronic conditions and diseases with long latencies. A sizeable disease burden in later life is likely to be directly attributable to chemicals exposures. We urge national and international agencies concerned with child labour and occupational health, to prioritize research and interventions aiming to reduce noxious chemical exposures in workplaces where children are likely to be present.

## 1. Introduction

### 1.1. Rationale

Hazardous child labour is defined by the 1999 ILO Convention No. 182 as: “work which, by its nature or the circumstances in which it is carried out, is likely to harm the health, safety or morals of children” [1]. Worldwide, there are over 218 million working children, of whom 73 million are child labourers involved in hazardous work that endangers their health and safety [2]. Child labour is most prevalent in Africa and the Asia-Pacific Region, where nearly 50% of child labours are 5–11 years old [2]. In both the Asia-Pacific region and in Latin America and the Caribbean, around 7.5% of children work as child labourers, however in Sub-Saharan Africa this number rises to over 20% [2]. Child labourers are often found in informal sector industries including agriculture, mining, manufacturing, domestic work, and construction [3,4]. They have minimal occupational safety and health (OSH)/workplace protections, given that it is illegal in many settings [5]. Children face varied occupational risks, including long working hours, injury risks from machinery or tools too big for them to handle, workplace violence, and low or no wages. Among occupational hazards, childhood exposure to chemicals can be especially pernicious, and a major threat to health [6].

**Chemical Hazards****in the Workplace.** Chemical hazards in the workplace can cause serious injury or illness to children, resulting in disability and chronic conditions, which can persist into adulthood [7]. However, evidence on longitudinal impacts of chemical hazards on child labourers is extremely scarce. Additionally, workplace exposure to hazardous chemicals is included in ILO guidelines as “work which should be prohibited” [1]. Whilst some chemicals are essential for protecting health, for example, as disinfectants or for protecting crops, they also pose serious health risks [8].

A wide range of chemicals can adversely affect health. Pesticides, heavy metals (e.g., lead and chromium), solvents (e.g., benzene), and banned persistent organic pollutants (POPs) are of particular concern [9]. Chemicals are used in numerous industries for different tasks. For the estimated 70% of child labourers employed in agriculture globally, work can involve handling, mixing and spraying pesticides [10]. Worryingly, children in low- and middle-income countries (LMIC) are often exposed to dangerous pesticides, which are banned in high-income countries (HIC) [11]. Around one million children work in mines, using explosives and chemicals like mercury [12]. Children working in manufacturing and trades, such as recycling and automobile repair are also at risk. For example, toxic lead fumes are released in lead recycling workshops, where often in LMIC, batteries are broken down manually with an axe [13]. Children working on rubbish dumps are exposed to a cocktail of metals, including cadmium, copper, and arsenic [14], as well as POPs [15] and gem industry workers use chromium to polish stones [16]. Examples of specific work tasks involving chemicals in different industries can be found in Table 1. POPs are of particular concern, as they are resistant to environmental degradation and their bioaccumulation has potentially adverse impacts on health and the environment. The Stockholm Convention (2001), signed and ratified by over 150 countries globally, aims to eliminate or restrict the use of the most hazardous POPs, such as polychlorinated biphenyl (PCBs) and dichlorodiphenyltrichloroethane (DDT).

Aside from work exposures, children may be exposed to hazardous chemicals through their environment, for example from living in a polluted area or by eating contaminated food. Research on environmental chemical exposures is more prevalent and health impacts include neurobehavioural deficits [17], early puberty and infertility [18,19], later life diabetes and obesity [20], increased respiratory symptoms [21], and immunotoxicity [22]. Childhood cancers, such as liver and thyroid cancers, have also been documented, alongside adult cancers from childhood chemical exposures, including breast and brain cancer [9].

**“Children Are not Little Adults”** [23]. Children and adolescents are particularly vulnerable to harmful workplace exposures and injuries [24]. Data from LMIC is limited, however injury rates among adolescents aged 15–17 in the USA were nearly twice that of workers 25 years and older [25]. Chemical exposures in childhood can cause severe disruption to the body’s systems, including the nervous, reproductive, endocrine, cardiovascular, immune, and respiratory systems [9]. Developing children are especially vulnerable and may have different susceptibilities during different life stages, especially during “critical windows of development” [23]. During these highly sensitive periods of development, children may be more at risk of adverse health outcomes, due to environmental exposures, than in less sensitive times [26]. For example, developmental exposures have been linked with neurobehavioural deficits, immune impairment, and precocious or delayed puberty [9]. These outcomes may be irreversible and have long-lasting impacts. In addition, children’s unique behaviours, physiology, and size place them at risk for absorption of higher chemical doses compared to adults [11]. 

Because of their age, child labourers often lack emotional maturity and are frequently illiterate and poorly educated [27]. Children are also susceptible to workplace violence and abuse and are less likely to be able to defend themselves relative to adults [28]. Additionally, child labour tends to be concentrated in particularly dangerous industries [10] and labour laws, including bans on child labour, are frequently not implemented in LMIC [29]. Examples of child vulnerabilities are shown in Figure 1.

Children’s absorption of anthropogenic and natural chemicals can occur by various pathways, such as inhalation, ingestion, dermal contact, or ocular exposure [30]. Exposures may be acute and high in dose or repeated, low-dose exposures over a long time period [9]. Poor quality work environments, for example with poor ventilation [31] and limited hygiene measures, such as infrequent handwashing [32], may increase exposure risk. Exposure factors related to personal characteristics, such as behaviour, age and socioeconomic status, will also determine an individual’s risk [9].

**Challenges for Chemical Risk Assessment among Children****.** Occupational Exposure Limits (OELs) have been developed by national OSH bodies for adults in some settings, however implementation and monitoring of these thresholds is limited in many LMIC, and they are not applicable for children [33]. Methods to monitor exposure among adults include workplace ambient air concentration monitoring (OEL) and biomonitoring of fluid or expired air (BEI) [34]. While the CDC originally established a blood lead threshold for children of 10 μg/dL, it has now been established that no blood level is safe [35]. Child-specific thresholds for other chemicals have not been officially established. Measuring chemical exposure levels in children also presents challenges. Exposure assessment in child labour studies is usually conducted using subjective self-report questionnaires. Biomonitoring methods, for example, using blood and urine, and environmental assessments of ambient air, soil, and water, are used less frequently. 

### 1.2. Objectives

While systematic reviews on child labour and health have been conducted [36,37,38,39], none offer detailed findings on the impact of chemical hazards on health. This review aims to synthesise and appraise the literature on health impacts of chemical exposures among child labourers in LMIC. LMIC were classified according to the World Bank’s definition of income status, using gross national income per capita [40]. The objectives are to:Conduct studies carried out between t chemical exposures, adverse health consequences, and the industries which put child labourers most at risk;Describe exposure and health outcome measurement issues;Identify the public health implications of chemical exposure among child labourers.

## 2. Materials and Methods

### 2.1. Search Strategy

This review followed PRISMA Guideline (Table A1) [41]. A pre-review protocol was developed and included the review aim and objectives, search strategy, selection criteria, and quality appraisal. However, the protocol was not formally registered, and selection criteria narrowed over the course of the review. The review protocol is available upon request from the lead author. A multi-stage systematic search was carried out between 25th and 27th July 2020 using four databases (MEDLINE, Embase, Global Health, and Scopus). The databases were selected for their comprehensive range of topics and relevance to the subject matter. The search strategy was designed with the assistance of the LSHTM librarian, using previous child labour and health systematic reviews [36,39] to inform keywords. The OSHA Occupational Chemical Database [42] was consulted to provide an overview of possible chemical exposures and their synonyms. Five key concept areas were identified for inclusion: “Child labour”, “Chemicals”, “Child”, “Occupational exposure” and “Industry”. A complete list of concepts, synonyms and MeSH terms are shown in Table A2.

Due to the number of irrelevant results produced during preliminary searches, search criteria were narrowed to include the synonyms for “child” as title searches only. Other search terms, for example chemical terms, were included as keywords. An example of the full electronic search strategy of MEDLINE is in Table A3. Backwards and forwards citation tracking of existing child labour and health reviews [36,38,39] was carried out to ensure that no important studies were missed.

### 2.2. Selection Criteria

Eligibility criteria were defined using the PICOS framework (Table 2).

The inclusion age group was chosen according to WHO criteria [9], which defines older children as being between 5 and 12 years of age and adolescents as 12–18 years of age. For the purposes of this review, we use the term “child labourers” to refer to working youth aged 18 or below.

Studies were excluded if they were commentaries, book chapters, case studies, editorials or if full texts could not be retrieved.

Search results were imported into reference manager, Mendeley, and deduplicated. Records were screened by title and by abstract, before the remaining full text papers were assessed for eligibility, by a single reviewer (NS). Due to the high number of potentially eligible studies meeting the inclusion criteria, the exclusion criteria were narrowed to focus the review to include:Studies carried out before the 1999 ILO “Worst Forms of Child Labour Convention (No.182)” [1]Studies carried out in HIC, for example United StatesStudies without a non-worker comparison group which included participants aged 18 or below

Studies carried out between the years of 2000 and 2020 and from LMIC were therefore only included.

### 2.3. Data Extraction and Analysis

Data extraction of included studies was performed using a standardized table with data entered and stored in Excel. Data extracted included: Author, title, year published, country, study type, age and gender of participants, sample sizes, type of control, industry, chemical exposure type, and health outcomes. Due to the heterogeneity of outcomes, it was not possible to do a meta-analysis and therefore a narrative synthesis was conducted.

#### 2.3.1. Exposure and Health Outcome Measurement

Four main chemical exposure types were identified and data was therefore synthesized according to these categories: (1) Pesticides; (2) Solvents; (3) Heavy metals; (4) POPs. Chemical exposures and health outcome exposures were classified as shown in Table 3. Definitions and methods of exposure measurement were determined by study authors. 

#### 2.3.2. Chemical Exposure Definitions and Measurement

The most basic measure of exposure is using sector as a proxy for exposure to hazardous chemicals, with no attempt made to empirically quantify specific task related exposures. Many studies use self-report questionnaires, which provided information on current work hours and work history, either as an inclusion criteria for participants or to ascertain the existence of dose-response relationships. As many subjects are illiterate, questionnaires are often administered by trained staff, sometimes with parental assistance. These provide some level proxy of exposure quantification, however more precise data is provided by biomarkers.

Biomarkers of exposure are commonly used to measure past and recent toxin exposure, with results linked with adverse health outcomes [43]. Biomarkers can include actual levels of the toxicant in fluids and tissues (e.g., blood lead levels), a metabolite of the toxicant, or an early reaction to the toxicant [43]. Biomarkers are often measured using samples of blood, hair, urine, and saliva. Whilst blood and urine chemical concentrations are often used for chemicals such as metals and POPs, commonly used biomarkers of pesticide exposure include inhibition of acetylcholinesterase (AChE) and butyrylcholinesterase (BChE) enzymes and increased levels of the urinary metabolite TCPy [44].

Environmental toxicant levels can also be used to characterise an individual’s exposure to chemicals [45]. These can be measured using air, water, soil, or food samples. Although measurements can be taken at an individual level, cruder measurements of the whole environment are more commonly used, due to issues with cost [46].

#### 2.3.3. Health Outcomes Definitions and Measurement

Health outcomes depend on the type of chemical, dose, and timing of exposure. High-dose exposures tend to produce obvious clinical effects associated with poisoning, for example, headache, dizziness, and convulsions, whilst low-dose, chronic exposures can lead to accumulation of toxins in living organisms [9]. 

Health impacts were defined as adverse health outcomes occurring at micro or macro level, ranging from DNA damage and abnormal biomarker readings, to diseases and pathologies affecting organs and whole-body systems. For the purpose of this review, health outcomes were defined by the study authors, to encompass the range of diseases and types of materials detected.

Aside from measuring exposure, biomarkers can also be used to measure health effects of chemical exposure [47]. In this review, biomarker findings are presented according to how the author described and used them. Again, the use of self-report questionnaires can be used to measure some health outcomes, for example, mental health outcomes may be assessed using POMS (Profile of Mood States) and the Children’s Anxiety Test. Various different measures are used for physiological tests, for example, lung function is measured with spirometry [48] and thyroid function looks at blood thyroxine (FT4) [49].

Neurobehavioural tests measure functions such as memory, concentration, and accuracy. Commonly used tests include the Behavioural Assessment and Research System (BARS) and the Wechsler Adult Intelligence Scale (WAIS).

Exposure to toxic chemicals can cause oxidative stress in cells. Cells under oxidative stress display different dysfunctions due to damage of their macromolecules, like protein, lipid, DNA, and RNA. This can lead to malignant cells [50] and DNA damage [51]. DNA damage is measured using a Comet Assay, as electrophoresis technique [51].

### 2.4. Quality Appraisal

Quality appraisal was conducted by one reviewer (NS) using the Critical Appraisal Skills Program (CASP) checklist [52] for cohort studies and the AXIS tool [53] for cross-sectional studies. Several domains related to internal and external validity are assessed by these tools, including sample size, sampling process, statistical methodology, and sources of bias. To aid the critical appraisal process, a points scoring system was developed for each criteria, which was applied in both tools (Yes = 2; Partial yes = 1; No = 0; Don’t know = 0—please see Figure A1 and Figure A2 for details). With predominantly cross-sectional studies included, studies were assessed according to the best methodological quality possible for this study type. The term “quality” was therefore used rather than “risk of bias” [54].

Study totals and percentages were calculated and used to determine overall study quality. A three-band scoring system (good quality ≥70%; medium quality 50–69%; low-quality < 50%) was devised in order to facilitate comparison between studies. These quality ratings were not used to exclude studies however, and detailed notes were also taken about individual study limitations on the limitations of each study. 

## 3. Results

The electronic search identified 5878 records with an additional three from reference lists (Figure 2). After removal of duplicates, 3898 were screened for inclusion first by title and then by abstract, leaving 306 full text articles to be assessed for eligibility. Of these, 272 were excluded for reasons such as having an environmental exposure or an adult study population. After applying the refined inclusion criteria to 34 papers, a total of 23 were selected for final inclusion.

### 3.1. Characteristics of Included Studies

Twenty-three papers from 20 studies were included in this review. Of the 20 studies, just one was a cohort study [55] while the rest were cross-sectional. One cross-sectional study [56] included measurements taken from two different study cohorts in 2005 and 2009. No qualitative studies meeting the inclusion criteria were identified. 

Studies were carried out in Bangladesh (*n* = 2), India (*n* = 1), Pakistan (*n* = 6), Lebanon (*n* = 2), Egypt (*n* = 5), Brazil (*n* = 1), Nicaragua (*n* = 1), Turkey (*n* = 1), and Indonesia/Zimbabwe (*n* = 1). The age range of participants was 6 to 18 years old, with some studies looking at males only (*n* = 9).

Studies looked at different industries, including agriculture (*n* = 5), mining (*n* = 1), waste disposal sites (*n* = 2), and different types of workshops (*n* = 12), including automobile repair, battery recycling, surgical instrument manufacturing, and gem polishing. Some studies looked at more than one industry type (*n* = 6).

Several studies stated that the working environment was below acceptable standards for ventilation, temperature, noise, water, and sanitation [16,57,58,59]. No PPE, such as gloves, goggles, or face shields, was worn in many studies [57,60]. In addition, bad hygiene habits, such as not washing hands, ingesting contaminated food, or smoking at work, were also reported [58].

### 3.2. Quality Appraisal

The one cohort study (and sole paper) was appraised to be medium quality. Of the 22 cross-sectional papers (reporting on findings from 19 studies), 6 were of high quality, 13 of medium, and 3 were low. No randomised control trials, the gold standard study type [61], were included. Most papers included appropriate study designs to address their specific objectives. The majority of studies gained ethical approval and consent from study participants. The main methodological flaws included lack of detail provided on sample sizes (power calculations and rationale) and sampling strategies, and no/limited information on non-response rates. A summary of the quality of the 22 cross-sectional papers is shown in Figure 3, with full results for all papers in Figure A1 and Figure A2. 

### 3.3. Evidence Synthesis of Health Outcomes by Chemical Type 

Key characteristics of included papers are in Table 4. Findings are reported by chemical exposure type: Pesticides (*n* = 5), solvents (*n* = 3), metals (*n* = 13), and POPs (*n* = 2), with synthesis of health effects for each chemical exposure in this section.

#### 3.3.1. Pesticides and Health Outcomes (*n* = 5)

Of the five pesticide papers included in the review, four looked at cotton crops in Egypt and the other at vegetables in Brazil. Three were medium quality, whilst two were good quality papers. All papers obtained self-report work history and biomarkers of exposure were also measured, including urinary TCPy [55,63] and plasma AChE/BChE [60,63]. Eckerman et al. [62] calculated an exposure index for each individual, based on the answers given by each participant in an interview about work history and current work status.

Urinary TCPy concentration was significantly higher in pesticide applicators than non-applicators in two papers [55,63]. However, differing results were found for AChE/BChE activity. Whilst one medium quality paper found no difference between applicator and non-applicator groups for AChE and BChE activity [63], two good quality papers reported significantly lower activity levels in applicator groups for AChE and BChE [56,60].

Two papers looking at neurological symptoms both reported significantly more symptoms in applicators, including dizziness and memory problems (Unadjusted Odds Ratios ranged from 1.18 to 15.3) [56,60]. Furthermore, depressed BChE activity was associated with increased symptoms [56].

Neurobehavioural testing used computerised and non-computerised methods. Tests were age appropriate and translated into relevant languages. Results for neurobehavioural tests were mixed, with applicators scoring significantly worse than non-applicators on all tests in one good quality paper, with effect sizes ranging from small (0.2) to large (0.8) [60]. Only memory and attention functions were significantly worse in another medium quality study however [63].

A dose–response relationship between AChE/BChE biomarker activity and neurobehavioural performance was found by three papers, though only for visual motor, memory, and perception functions [56,60,62]. Conversely, Rohlman et al. [63] did not find evidence of a dose–response relationship, possibly due to the small sample size and the moderate level of pesticide exposure. The fact that Ismail et al.’s [56] results were confirmed by testing in two different cohorts, adds validity to results. However, the “exposure index” used by Eckerman et al. [62], although individual, was unvalidated and derived from a small number of participants. Overall, results suggest that some (but not all) aspects of neurobehavioural development are affected by level of pesticide exposure. 

With regard to pulmonary function, the medium quality cohort study by Callahan et al. [55] found no significant difference between groups for self-reported wheeze and percent predicted forced expiratory volume (FEV1) and forced vital capacity (FVC). Cumulative TCPy, which was measured eight times over ten months, was inversely associated with percent predicted FEV1 and FVC at day 146 (at the end of the application season, when TCPy was elevated), but not day 269 (TCPy returned to normal levels). This suggests that urinary TCPy is inversely associated with lung function independent of applicator status. 

Whilst all papers looked primarily at exposure to the OP pesticide chlorpyrifos, application cycles involved other pesticides, for example carbamates, which may have influenced results. 

#### 3.3.2. Solvents and Health Outcomes (*n* = 3)

Three papers of one Lebanese study looked at solvent exposure in automotive spray painting, mechanical repair, and furniture painting workshops [64,65,66]. Six solvents were identified and assessed including benzene and toluene. Solvent-exposed working children were compared to non-exposed children. Exposure status in two medium quality papers [65,66] was assessed using self-reported work history, however one good quality paper [64] used personal ambient air levels measurements to calculate individual exposures for each participant. This involved using monitors clipped onto each child’s clothing, which recorded individual exposure levels to solvents in the working day. 

Health outcomes for solvent studies included neurological symptoms, neurobehavioural function and mood. Solvent-exposed children had significantly more symptoms than the two age matched referent groups of non-exposed working children and non-working school children. Symptoms included light-headedness, memory deficits, poor concentration, and headaches [65,66].

Solvent-exposed children also scored worse on computerised and manual non-computerised neurobehavioural function tests. Working-exposed children had a lower education level and 77% self-reported as having poor literacy skills and this could therefore have confounded results. Despite this, significant differences remained even after adjusting for education. Reaction speed, motor dexterity, and memory were all significantly worse in the solvent-exposed group compared to controls, but there was no difference for accuracy [65,66].

One good quality paper used the individual solvent exposure measures to look at dose–response effects [64]. Children with higher cumulative exposures reported more neurotoxic symptoms (*p* = 0.02) and performed worse on neurobehavioural tests for reaction time, perception, sustained attention, memory, and motor coordination. 

With regard to mood, exposed working children were significantly angrier, were more confused, and had higher irritability [65,66]. Interestingly, there was no difference between low and high exposure groups, which indicates that there was no dose–response relationship detected [64]. 

Selection bias may be an issue for this study, as 18% of workplaces approached declined to participate. The results therefore lack external validity. 

#### 3.3.3. Metals and Health Outcomes (*n* = 13)

Thirteen papers, related to thirteen separate studies, looked at metal exposures across industries including auto spray painting, mechanical repair, battery recycling, carpet weaving, gold mining, and pottery workshops. Papers looked specifically at lead (*n* = 6), mercury (*n* = 1), chromium (*n* = 1), lead and cadmium (*n* = 1), and also a mix of metals (*n* = 4). Study quality ranged from good (*n* = 3) to medium (*n* = 7) and low (*n* = 3).

Exposure was measured in a variety of different ways, including: child labourer exposed to chemicals only (*n* = 2), self-reported work history only (*n* = 5), self-reported work history plus minimum blood metal concentration (*n* = 1), self-reported work history plus environmental assessment (*n* = 1), child labourer exposed to chemicals plus environmental assessment (*n* = 2), blood metal concentration (*n* = 1), and an exposure equation for inhalation, ingestion, and dermal contact (*n* = 1).

Health outcome measures included blood, hair and urine concentrations, oxidative stress and DNA damage, neurobehavioural tests, pulmonary, thyroid and liver function tests, and mental health. Papers looking at metal concentrations in blood, hair, and urine samples, found significantly higher levels in exposed groups compared to unexposed groups for blood, hair and urine mercury concentrations [71], blood lead [13,58,67,68,70] hair lead [57,58], blood and hair cadmium [58], urinary nickel [59], and urinary chromium [59]. Importantly, in a high quality paper, urinary chromium concentrations were 35 times higher in working children and largely in excess of the occupational BEI for adult workers [75]. Conversely, one low quality paper reported no significant difference for blood chromium levels between gem industry workers and non-workers [16]. Numerous limitations were found for this study methodology however, including a poor sampling process, small sample size, and insufficient reporting of methods. Another good quality paper [68] found that 100% of workers in a variety of different workshops, including auto repair, pottery production, and smelters, had blood lead levels higher than CDC’s original 10 μg/dL threshold level. Studies of mixed metals reported increased concentrations for exposed workers in blood, hair, saliva [72], and urine [72,73]. Hair was found to have the highest bioaccumulation of metals [72].

Three papers assessed oxidative stress and DNA damage caused by lead exposure [69] or a combination of metals [14,59]. All three reported significantly altered oxidative stress parameters in exposed groups, with two [14,69] reporting increased DNA damage. Specifically, Lahiry et al.’s low quality paper stated that one marker of DNA damage was 15.6 times higher in the exposed group, although limitations in study methodology must be considered [14]. One paper also found that those exposed had decreased body weight compared to controls [69]. 

Only one medium quality paper looked at the impacts of mercury exposure in gold mining workers [71]. Outcomes were compared between exposed workers, those exposed by living near the mines, and a non-exposed control group. Exposed participants had symptoms of mercury intoxication, including ataxia and deceased reflexes, and also performed worse in two neurobehavioural tests looking at concentration, intentional tremor and coordination. No differences were found between groups for memory and visual-motoric capacity tests. 

Other health outcome assessed due to metal exposures included pulmonary, thyroid and liver function, and blood pressure. Workers in surgical instrument manufacturing reported a significantly higher prevalence of asthma and dry cough (*p* = 0.02), but no difference in pulmonary function [59]. The paper by Dundar et al. [70], which assessed the impacts of lead from auto repair workshops on thyroid function, revealed that long-term, low level lead exposure may lead to reduced thyroid function, even at low exposure levels. It found that exposed workers had significantly lower FT4, a commonly used measure of thyroid function, and that blood lead was negatively correlated to FT4 levels. Other thyroid function measurements and also thyroid volumes did not differ between groups, however. No differences were also reported for liver function [14] and blood pressure [59]. 

Only one good quality paper looked at mental health [67]. No differences were found for workers compared to non-workers for anxiety, helplessness, and self-esteem, despite a third reporting dissatisfaction with their jobs. The paper reported that this lack of difference may have been due to study participants being male and therefore less likely to report mental health symptoms. Another paper found that working children were more likely to have adverse habits, such as smoking and drinking [68].

Only two papers compared blood lead levels between different workshop types. In one, blood lead was higher in child labourers from pottery workshops, compared to those working in automobile repair, car batteries, smelters, radiators, and garbage collection [68]. The other found higher blood lead levels in battery recyclers compared to welders [58]. No other studies were found comparing exposures in different workshop types.

#### 3.3.4. POPs and Health Outcomes (*n* = 2)

Two medium quality papers looked at the results of one study conducted in 2002 at a waste disposal site in Nicaragua [15,74]. Exposure status was measured by a self-report work history questionnaire. Five different study groups were investigated including, workers living at the site, workers living nearby, and non-workers living in different locations. The influence of fish consumption was also evaluated in the children and in groups of 15–44 year old women.

Serum POP concentrations were measured, with Cuadra et al. [74] looking at different POPs, including DDT and PCB, and Athanasiadou et al. [15] focusing on PBDE only. Children working in waste disposal had higher serum POP levels than non-working reference groups in this study. Indeed, children working and living at the site had 20–50 times higher PBDE levels than three referent groups living in other regions [15]. Exposure through inhalation was reported to be of greater magnitude than dietary exposure, although POP levels were slightly higher in fish eaters than not [15,74]. Study limitations included small sample sizes, and the fact that within group variation was not calculated due to the use of pooled blood samples. 

#### 3.3.5. Health Outcomes Not Impacted by Chemicals 

Papers examining the relationship between chemical exposure and pulmonary function [55,59], thyroid volume [70], blood pressure [59] and liver function [14] among child labourers provided no significant associations. However, the fact that each of these health impacts were only investigated in a single study makes it impossible to draw specific conclusions. 

## 4. Discussion

### 4.1. Key Findings 

Exposure to hazardous chemicals among child labourers clearly constitutes a major public health challenge. This review found adverse health impacts for all chemical types. Pesticides, metals, and solvents were associated with neurobehavioural deficits and neurological symptoms, solvents and metals with mental health issues and metals with oxidative stress, DNA damage, poor growth, asthma, and hypothyroidism. All chemical types, including POPs were linked to increased blood, hair, and urine chemical concentrations or other abnormal biomarkers. An overview of the reviewed papers and the health impacts of the different chemicals is shown in Figure 4.

Working children in the studies were often under the age of ten and had very poor literacy skills [66]. Safety regulations were rarely implemented and simple hygiene measures were not followed [58]. Child labourers working in agriculture and mining, on waste disposal sites and in trades, like battery recycling and pottery production, appear particularly at risk. Indeed, lead battery manufacturing and recycling are now the most significant sources of lead exposure in the world [76]. 

The results build on previous findings in child labour and health systematic reviews, however, previous reviews were broad and did not focus on chemical exposures, which can be especially pernicious for children [36,39]. Our results for lead specifically concur with a recent UNICEF report on childhood lead exposure, which found that exposure to lead can damage children’s developing brains, nervous system, and lungs and even low levels of lead can cause significant damage among exposed children [77]. The discussion focuses on public health implications of selected findings and measurement issues identified from this review. 

### 4.2. Public Health Implications

Dangerous levels of chemicals were repeatedly found in blood, hair and urine samples, with some studies reporting exposures way above adult OEL and BEI thresholds [59,68]. No specific thresholds for children were reported. Extensive evidence has documented the numerous health impacts associated with chemical exposures, including cancers and organ damage:Cancer/malignancy—work-related cancers are the largest cause of workplace mortality in adults and account for 32% of deaths [78]. High levels of chromium, lead, and cadmium were identified in biomatrix samples from the working children in this review. This is of particular concern, as the IARC (2012) has classified cadmium and chromium as carcinogenic to humans and lead as probably carcinogenic [79]. Indeed, chromium is a recognised carcinogen [80] and has been linked with lung cancer [81] and increasing blood lead level has been associated with increased trends for lung and brain cancer [82]. Many pesticides have also been classified by IARC as carcinogenic or probably carcinogenic to humans, with cancer excesses observed for numerous cancers, including prostate, leukaemia, thyroid, and testicular cancer [83].Other health outcomes—studies of adult populations have linked lead with kidney and brain damage [84] and cadmium with irreversible kidney and lung damage [85]. Chromium has been associated with immunosuppression [81]. Elevated mercury levels were found in the child labourers included in this review. This is significant when considering that mercury can produce harmful effects on numerous body systems, including the nervous, digestive, and immune systems, as well as the lungs and kidneys [86]. Elevated blood POPs, such as PCBs, have been identified as endocrine disrupting chemicals and have been implicated in numerous reproductive disorders [87].

Neurobehavioural defects and neurological symptoms were linked with both pesticide and solvent exposure among child labourers in this review [56,60,62,63,64,65,66]. Importantly, a dose–response relationship was demonstrated between solvent or pesticide exposure and neurobehavioural deficit [56,60,62,63,64]. Similar results were found in another large-scale review, examining neurodevelopmental effects in children exposed to organophosphate pesticides in childhood [47]. Impacts could be extensive for child labourers who work long hours and have years of cumulative exposure to chemicals. In the short-term, neurobehavioural deficits are likely to interfere with a worker’s capacity to respond quickly and safely to new demands and to learn new information, thereby increasing injury risk [66]. Longer-term however, deficits gained in childhood may also be sustained into adulthood, which will be detrimental to employment opportunities and quality of life in adulthood. Crucially, no studies were found investigating child labourer exposure to toxic metals and neurobehavioural deficits. This is an important research gap when considering that increased blood lead is associated with IQ deficits in children [77]

Child labour was linked to a high prevalence of mental and behavioural disorders in two previous reviews [38,88]. However, few studies in this review investigated chemical exposures and mental health disorders in child labourers. No studies were found looking at specific mental health conditions, such as depression. This is important, as childhood incidence of anxiety and depression has been linked to an increased risk of mental health disorders in adulthood [89] and pesticide exposure has been linked with suicide among adult workers [90].

Findings from this review on metal exposure and cellular oxidative stress are echoed in research with adults. Adult studies have reported DNA damage in petrol pump workers [91] and workers exposed to electronic waste [92]. These findings have serious public health implications, when bearing in mind that oxidative stress has been associated with malignant cells [50], atherosclerosis [93], aging [94], and neuro-degenerative diseases in adults [14].

### 4.3. A sizeable Burden of Disease in Later Life

Childhood exposure to chemicals has been linked to disease incidence in adulthood in many studies. For example, childhood exposure to pesticides has been linked to adult-onset rheumatoid arthritis [95]. It is likely that a sizeable disease burden in later life will be directly attributable to childhood occupational exposure to chemicals. In LMIC where child labour is especially prevalent, the health and welfare costs, as well as the potential impact on lifetime income levels, may be substantial [10]. For example, a study of the economic costs of lead exposure to child labourers found that a 1 μg/dL of blood lead concentration is associated with a 0.25-point decrease in IQ and the loss of 1 IQ point corresponds to an overall reduction of lifetime earnings of 2.4% [96].

Toxins may also be passed on to the children of current child labourers. Indeed, a survey conducted in Managua found that 21% of adolescent females were pregnant or already mothers [74]. This means that future generations of children may be impacted by chemicals exposures from their parents.

Although in an ideal world, hazardous child labour should not exist, this complex societal issue does not have easy policy solutions. Banning child labour can have adverse consequences, for example, increased child labour prevalence [97]. It became costlier for employers to hire children in India, which reduced child wages, prompting families to send more children out to work [97]. In Brazil, banning child labour had differential effects, with non-white youth being less likely to be employed or have a formal job later in life [98]. Furthermore, child labour bans can prompt shifts of children from legitimate work to jobs in the illegal economy, where they may be subject to worse abuses and exploitation [5]. Child labour is often essential for economic survival and can provide some benefits, including valuable life skills, such as resilience and adaptability [99]. Indeed, working children themselves cite positive impacts of work, such as learning valuable trade and agricultural skills, alongside communication and problem solving skills in less hazardous forms of work [100]. While children’s views are important, we should be cautious that children’s positive evaluation of their occupations can reflect personal and cultural investment in coping with familiar situations, even when they are hazardous and exploitative [101]. The debate about appropriate child labour policy responses is complex, requiring a balance between protecting children’s health while assuring family income [102]. Sustained cash transfers and targeted micro-insurance are likely necessary complements to legislative interventions [103]. Furthermore, most social protection interventions are focussed on preventing initial entry to child labour or shifting children to full-time school immediately. These are not options for scores of children without good quality education options nearby, or for whom returns to formal education in labour markets is low [103]. Very few transitional education models exist for child labourers [104], with limited intervention evaluations. Yet these are the very interventions that can offer hope for the 218 million children who are currently working worldwide. Furthermore, occupational health interventions (discussed below) could be implemented to assist currently working children, which could immediately and positively impact their health.

### 4.4. Measurement Challenges in Exposure and Outcome Assessments

#### 4.4.1. Validity and Reliability of Self-Report Questionnaires for Measuring Exposures Is Unclear

Self-report questionnaires were used by many studies for both exposure and health outcome measurement. However, the validity and reliability of self-reports is questionable. Child labourers usually have poor literacy skills and have difficulty understanding recall periods and understanding chemical terms such as “pesticides” [59]. Whilst some studies reported using trained staff, translated questionnaires, and modified interview questions to suit the study populations, nearly half the studies did not clarify how self-report questionnaires were administered. In some cases, parental assistance was used, and this may also have influenced results. It is therefore possible that work histories and health outcomes may have been misreported, due to issues such as age, literacy skills, recall bias, and acquiescent bias [105]. In addition, validation of child OSH self-report questionnaires, via cognitive interviewing, is needed particularly with different age groups [106]. For example, children as young as five may be able to provide self-reports of pain, though only older children can understand concepts such as self-esteem [105]. Self-reports could also be validated by adding an OSH observational component to studies, with OSH researchers following child labourers at work to record chemical exposures observed, using standardized assessment forms and procedures [107].

Several studies in this review used job title as the only means to classify exposure to chemicals. However, using job titles as indicators of occupational chemical exposure can lead to exposure misclassification, as exposure will vary according to the specific tasks conducted within that role [108]. 

#### 4.4.2. Single Biomarker Assessment Is Inadequate for Accurate Measurement 

Whilst biomarkers can be used to indicate that chemicals have been absorbed into the body, they cannot provide information on the course, route or duration of exposure [109]. Importantly, as only a single measure was taken in the majority of studies, it was not possible to differentiate between low-level chronic exposures and high-level short-term exposure [110].

Whilst most child labour studies used traditional blood and urine biomatrices, it should be noted that these only reflect recent exposures and using them to measure chronic exposures can cause bias via underestimation of effect sizes [111]. Recent reports of human biomonitoring in general have highlighted the benefits of using non-invasive biomatrices, such as hair and saliva [112]. These are easy to collect, store, and transport [19] and hair can reflect past exposures up to a year [111]. However, hair may be susceptible to contamination, for example due to diet and washing, which should be considered when using this type of sample [19].

In line with evidence from adult studies [44], pesticide studies showed significantly decreased cholinesterase activity in pesticide applicators compared to controls. Due to individual variability, two biomarker measurements are needed, including one at baseline [60]. Yet studies in this review only measured cholinesterase activity once, reflecting exposure at that particular point only. This is problematic as normal cholinesterase levels can vary by as much as 300% [113]. Validity of results must therefore be questioned due to non-differential misclassification. At least two measurements of biomarkers are necessary for public health monitoring of chemical exposures.

#### 4.4.3. Workplace Chemical Exposure Is Impossible to Isolate from Wider Environmental Exposures 

It is not possible to attribute adverse health effects from chemicals solely to those obtained through work. Indeed, child labourers will be exposed to chemicals from other sources on a daily basis. Children living in agricultural areas may be at increased risk of exposure due to the location of their homes and “take home” exposures from their parents [114], as found in this review [63]. Likewise, diet can contribute when metals are present in the environment, as was found with fish consumption in this review [71] and increased POPs [15,74]. Traffic and industrial pollution also present chemical risks [73] and transfer from mother to foetus in utero [115]. However, this review found that where comparisons were made between workers and non-workers living close to a work site, adverse health impacts were usually greater for workers [15,68,71,74]. As for biomarker monitoring, studies in this review only took environmental measurements, for example of soil, water, and air toxins, at one point in time. This is problematic, as chemicals are not distributed evenly in the environment [45]. 

### 4.5. Limitations of This Review

The majority of studies in this review were carried out in Pakistan, Lebanon, and Egypt, with only two studies in Latin America, one in Africa and no studies from China. This limits generalisability of finding to other locations. Four out of the five pesticide studies were carried out in Egypt, despite agriculture being the most common child labour sector globally. Due to time constraints, studies from high-income countries were excluded. Recent reviews have covered occupational health of young workers in HICs [29,116]. 

A limitation of this review is that some potentially useful databases were not searched, e.g., chemical database Haz-Map [117] and CISDOC, an occupational health database [118]. Studies were limited to the English language, so important studies may have been missed, for example, from Brazil where child labour and health research is common. Citation tracking may have helped overcome these limitations. A further limitation was that one author conducted full quality appraisal of studies. Extensive discussion and random checking of quality scores was conducted by the second author, to mitigate against potential bias.

The review included mainly cross-sectional studies, which therefore do not include empirical evidence on impacts of chemical exposures in adulthood. Section 4.3. indicates the potential disease burden in later life based on environmental exposure studies.

Limitations of the quality appraisal process should also be noted. The RoB-SPEO [119], a specialised tool for occupational risk factor studies, was not used due to its specific focus on prevalence of exposures. The AXIS tool [53] was chosen because it is a multi-disciplinary tool, specifically designed for cross-sectional studies. As with all such tools however, a degree of subjective assessment is required, and this may bias results. The numerical scale devised by the author, although not validated for use in this study, may have helped ensure comparability across studies.

However, a major strength of this review is that all studies included had at least one control group, usually non-working children. This review may be the first to synthesize impacts of hazardous chemicals on child labourers.

### 4.6. Priority Research Recommendations

We found no longitudinal studies of chemical exposures among child labours, including studies of adults who worked as child labourers. Longitudinal studies are needed, to examine long-term health effects, especially for conditions with long latency periods. 

Cohort and case control studies are also needed, using methods such as biomonitoring and other clinical measures, rather than self-report questionnaires. Qualitative research may also provide a broader understanding of the impacts of working with chemicals from the perspective of child labourers themselves. In addition, a systematic literature review is needed looking specifically at potential public health interventions to prevent and limit adverse health consequences from chemical exposures among child labourers. 

Specific areas of future research are identified in Table 5.

### 4.7. Priority Interventions/Practices

Interventions to reduce chemical exposure among child labourers were beyond the scope of this review. However, we offer ideas for public health interventions that would benefit from further exploration, by public health agencies and OSH bodies in LMIC: 

*Developing chemical hazard thresholds*. Development of child toxin thresholds, similar to OEL and BEI, should be a priority. These thresholds should consider acute toxic effects, as well as prolonged low-level exposures [70]. A number of studies referred to adult toxic thresholds [59,68], however it is likely that child thresholds are much lower than these. For example, there is no safe dose of childhood lead exposure [122]. 

*Conducting OSH inspections in high-risk sectors.* Battery recycling and agricultural roles with pesticides are particularly high risk for adverse health effects. Inspections are recommended in these areas, including environmental assessments and monitoring of relevant biomarkers in child labourers. At least two measurements should be taken to ensure that variability is taken into account. Issues exist with the practicalities of this recommendation, however, as it is likely that worksites would be averse to any type of monitoring and attempting to do this might drive child labourers even further underground. The ethics of medical testing on children must also be considered. Non-invasive testing, for example using hair, may be more appropriate for children, whilst also providing a more reliable estimate of chronic exposure. 

## 5. Conclusions

Chemical exposures have devastating health consequences for child labourers. Neurobehavioural deficits, mental health disorders, DNA damage, poor growth, thyroid issues, and abnormal biomarkers were pervasive in this review. Children are especially vulnerable to chemical exposures, due to their physiology and behaviours, yet workplace protections are limited and safety regulations are rarely implemented. Research is urgently needed to discern long-term health impacts, as well as research on effective interventions to reduce chemical exposures in workplaces where children are present. This review, combined with evidence from adult studies and those focusing on environmental exposures, hints at the possible scale of the problem. A sizeable disease burden in later life is likely to be directly attributable to chemicals exposures. On an individual level, hazardous child labour involving chemicals limits the development, welfare, and health of children, with consequences lasting into adulthood. When considering the large numbers of child labourers likely to be exposed to chemicals, the long-term public health implications will be substantial. We urge national and international agencies concerned with child labour and occupational health, to prioritize research and interventions aiming to reduce noxious chemical exposures in workplaces where children are likely to be present.

## Figures and Tables

**Figure 1 ijerph-18-05496-f001:**
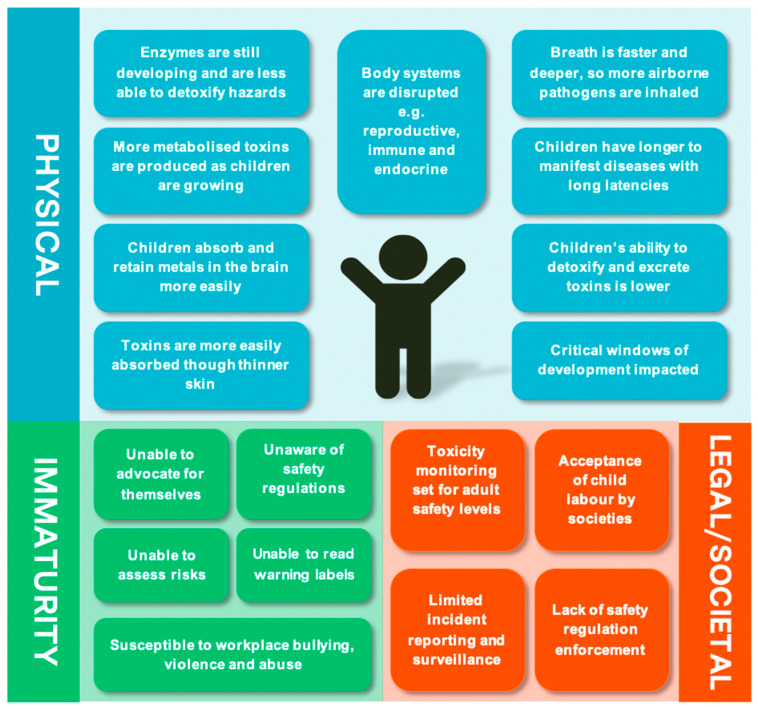
Vulnerabilities of child labourers to chemical exposures. Source: Developed by authors, based on articles by the ILO [3]; WHO [9] and Sámano-Ríos et al. [29].

**Figure 2 ijerph-18-05496-f002:**
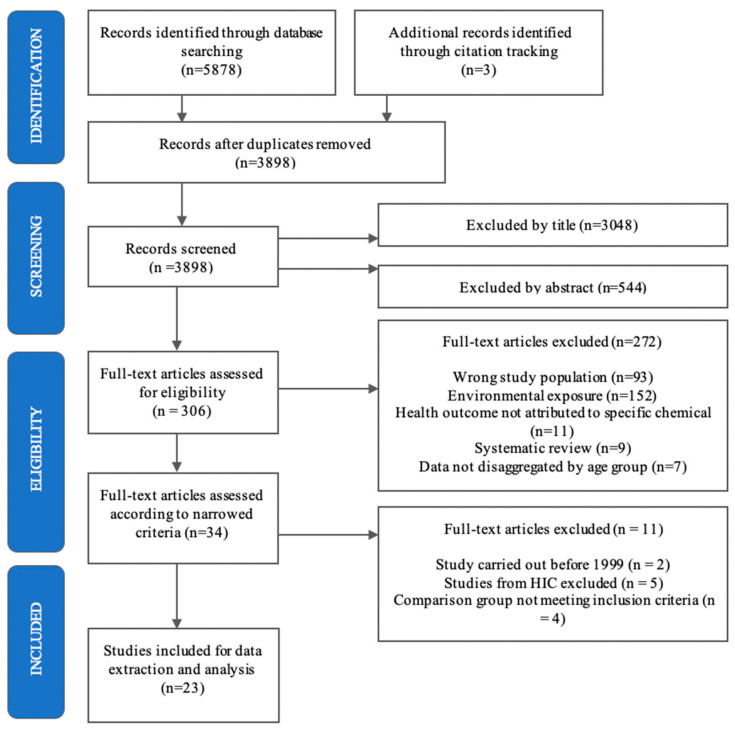
Prisma flow chart of study selection.

**Figure 3 ijerph-18-05496-f003:**
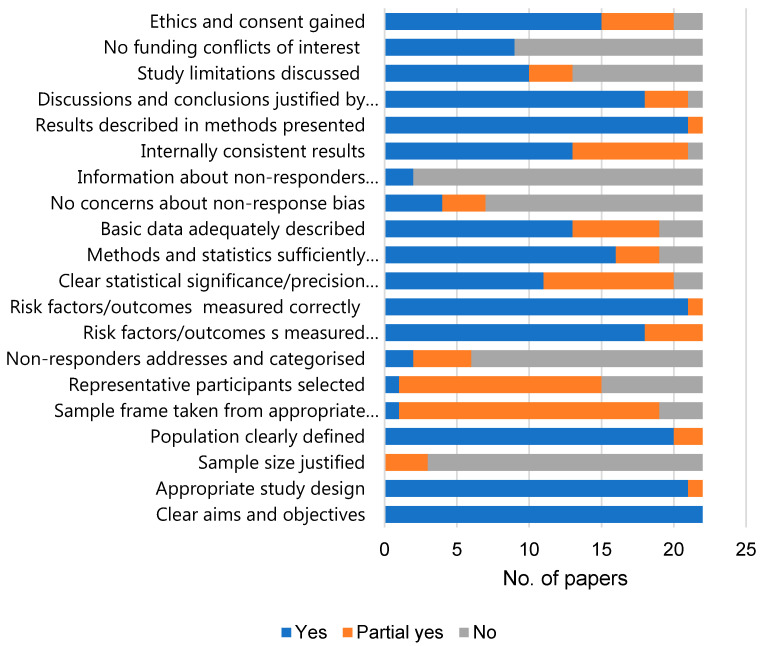
Quality appraisal results for cross-sectional papers. (cohort study by Callahan et al. [55] not included).

**Figure 4 ijerph-18-05496-f004:**
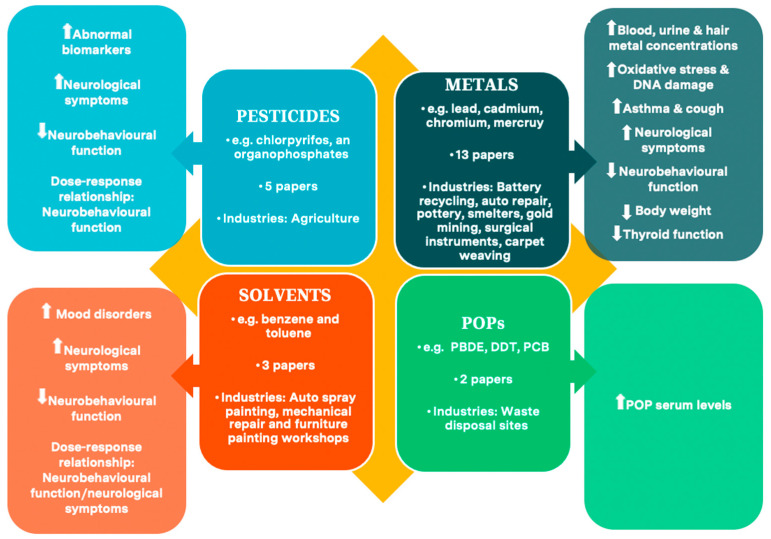
Overview of review studies and main heath impacts for exposed children.

**Table 1 ijerph-18-05496-t001:** Examples of chemicals in different industries.

Types of Chemicals	Examples of Chemicals Commonly Used by Child Labourers	Types of Industries Using These Chemicals	Specific Tasks Conducted by Child Labourers
Pesticides	Organochlorides e.g., dichlorodiphenyltrichloroethane (DDT) and chlorinated alicyclics Organophosphates e.g., chlorpyrifos, parathion Carbamate insecticides e.g., aldicarb, carbofuran	Agriculture, plantations, other rural sectors	Handling, mixing, spraying pesticides
Heavy metals	Lead, mercury, chromium, cadmium	Mining and quarrying, construction, service sector and street work, manufacturing, textiles, leather, footwear	Mining tasks (e.g., crushing and amalgamating, sieving, washing, and sorting), vehicle repair, trash recycling, shoe-shining, making textiles, tanning leather, ceramics
Solvents	Isopropanol, benzene, toluene, xylene, and solvent mixtures, such as white spirits.	Food and drink, construction, printing, plastics, rubber, textiles, leather, footwear, manufacturing, dry cleaning	Making textiles, tanning leather, painting, printing, plastic product works, pressing clothes, cleaning
POPs *	DDT, polychlorinated biphenyl (PCB), polybrominated diphenyl ethers (PBDE)	Agriculture, vehicle repair, plastics, construction, electronics	Pesticide use, painting, rubbish recycling, waste incineration

* Persistent organic pollutant (POP).

**Table 2 ijerph-18-05496-t002:** Inclusion and exclusion criteria.

Topic	Inclusion	Exclusion
Participants: Age	Participants are ≤18 years	Participants are >18 years old. Studies include those who are 18 years and under, however without disaggregated data for this age group
Participants: Workers	Participants are child labours, as defined by the study authors	Studies which do not involve work (e.g., those which are limited to environmental exposures only)
Interventions/exposures	A clearly defined chemical exposure	No specific chemical exposure
Comparisons	No criteria defined	No criteria defined
Outcomes	Health outcome directly related to occupational chemical exposure Acute or long-term outcome	Outcome not disaggregated according to exposure type
Study design	Qualitative studies Quantitative studies (intervention/case control/cohort/cross-sectional)	Grey literature
Language	English language only	Non-English language

**Table 3 ijerph-18-05496-t003:** Chemical exposures and health outcome classifications.

**Author-Defined Exposure**	**Exposure Measurement**
Child labour in the presence of chemicals	Sector where chemicals are used (usually based on questionnaire. Sometimes sector is used as a blanket category of chemical exposure by researchers before the study)
Current work status/work history	Self-report questionnaire to determine number of hours at workplace where chemicals are used and number of years in sector
Scientific measurement using specialised equipment	Biomarkers of exposure (e.g., chemical concentrations or metabolite levels) Environmental assessment to measure workplace exposure to chemical levels (e.g., environmental air, water, soil, or food samples) *
**Health Outcome Types**	**Examples**
DNA damage	Oxidative stress and DNA damage
Biomarkers of effect	Toxin and metabolite levels (measured in blood, urine, hair, and saliva)
Organ e.g., lung, heart, liver, skin, kidneys	Cancer, diabetes, asthma, kidney disease, dermatitis
Body system e.g., Cardiovascular, endocrine, respiratory, neurological, reproductive, immune	Cardiovascular disease, neurotoxic symptoms, neurobehavioural deficits, hypothyroidism, hypertension, pulmonary function
Mental health	Mood disorders e.g., anxiety, depression
Non-specific symptoms	Wheezing, nail discolouration, fatigue

* in Table 4, “environmental exposure” is used to describe studies where authors use these measurements as exposures, while “environmental assessment” is used to describe studies where authors use measurements as an outcome.

**Table 4 ijerph-18-05496-t004:** Key characteristics of reviewed papers.

Author Country (Year)	Chemical	Industry	Participant Age	Participant Gender	Author-Defined Sample Description	Author-Defined Exposure Measurement	Author-Defined Health Outcomes	Paper Quality
Ismail et al. Egypt (2017) [56]	Pesticides: Chlorpyrifos	Agriculture: Cotton	12–18 years	M	Pesticide applicators 2005 (*n* = 41); 2009 (*n* = 21) Non-applicators 2005 (*n* = 38); 2009 (*n* = 20)	SRQ: Current work status/work history/non-work chemical exposures	Neurobehavioural tests-BARS/WAIS-R Neurological symptoms Plasma BChE	Good
Abdel Rasoul et al. Egypt (2008) [60]	Pesticides: Chlorpyrifos	Agriculture: Cotton	2 groups: 9–15 years 16–18 years	M	9–15 years Pesticide applicators (*n* = 30) Non-applicators (*n* = 30) 16–18 years Pesticide applicators (*n* = 20) Non-applicators (*n* = 20)	SRQ: Current work status/work history BIO: Plasma AChE	Neurobehavioural tests-WAIS Neurological symptoms	Good
Eckerman et al. Brazil (2007) [62]	Pesticides: Unspecified OP	Agriculture: Green vegetables	10–18 years	M/F	Farmworker schoolchildren (*n* = 38) Urban schoolchildren (*n* = 28)	SRQ: Current work status/work history ENV: Non-work chemical exposures-Exposure index	Neurobehavioural tests-BARS	Medium
Callahan et al. Egypt * (2014) [55]	Pesticides: Chlorpyrifos	Agriculture: Cotton	12–19 years **	M	Pesticide applicators (*n* = 38) Non-applicators (*n* = 24)	SRQ: Current work status/ non-work chemical exposures) BIO: Urinary TCPy	Self-reported wheeze Lung function	Medium
Rohlman et al. Egypt (2014) [63]	Pesticides: Chlorpyrifos	Agriculture: Cotton	12–18 years	M/F	Pesticide applicators (*n* = 21) Non applicators (*n* = 20)	SRQ: Current work status/work history BIO: Urinary TCPy. Plasma AChE/BChE	Neurobehavioural tests-BARS/WAIS	Medium
Saddik at al Lebanon (2009) [64] ***	Solvents	Auto spray painting, mechanical repair and furniture painting workshops	10–17 years	M	Solvent-exposed workers (*n* = 100) Non-exposed workers (*n* = 100) Non-working, non-exposed schoolchildren (*n* = 100)	SRQ: Current work status/work history ENV: Ambient air levels of six solvents	Neurobehavioural tests-PIPS/non-computerised tests Mood-POMS	Good
Saddik at al Lebanon (2003) [65] ***	Solvents	Auto spray painting, mechanical repair, furniture painting workshops	10–17 years	M	Solvent-exposed workers (*n* = 100) Non-exposed workers (*n* = 100) Non-working, non-exposed schoolchildren (*n* = 100)	SRQ: Current work status/work history	Neurological symptoms Neurobehavioural tests-PIPS	Medium
Saddik at al Lebanon (2005) [66] ***	Solvents	Auto spray painting, mechanical repair and furniture painting workshops	10–17 years	M	Solvent-exposed workers (*n* = 100) Non-exposed workers (*n* = 100) Non-working, non-exposed schoolchildren (*n* = 100)	SRQ: Current work status/work history	Neurological symptoms Neurobehavioural tests-non computerised tests Mood-POMS	Medium
Nuwayhid et al. Lebanon (2005) [67]	Metals: Lead	Carpentry, mechanics, metal works	10–17 years	M	Workers (*n* = 78) Non-workers (*n* = 60)	SRQ: Work history	Mental health Physical exam Blood lead Haemoglobin/ferritin	Good
Moawad et al. Egypt (2015) [68]	Metals: Lead	Auto repair, car batteries, smelters, radiators, pottery workshops, garbage collection	6–18 years	M/F	Non-workers with moderate living standard (*n* = 100) Non-workers in slums (*n* = 100) Schoolchildren (suburban *n* = 70; urban *n* = 30) Workshop group (*n* = 100)	SRQ: CL status ENV: Water, dust, soil *****	Blood lead Haemoglobin	Good
Sughis et al. Pakistan (2012) [59]	Metals: Various	Surgical instrument manufacture	10–14 years	M	Exposed workers (*n* = 104) Schoolchildren (*n* = 75)	BIO: Urine chromium and nickel concentrations	Respiratory symptoms Lung function-spirometry Blood pressure Oxidative DNA damage	Good
Arif et al. Bangladesh (2018) [69]	Metals: Lead	Battery recycling	10–14 years	M	Exposed workers (*n* = 30) Non-exposed workers	SRQ: Current work status/work history BIO: Minimum blood lead concentration	Oxidative stress DNA damage-Comet assay Growth retardation	Medium
Shah et al. Pakistan (2012) [13]	Metals: Lead	Battery recycling	12–15 years	M	Exposed workers (*n* = 118) Exposed non-workers (*n* = 89) Non-exposed (*n* = 95)	SRQ: Current work status/work history	Blood lead	Medium
Dundar et al. Turkey (2005) [70]	Metals: Lead	Auto repair	15–17 years	M	Exposed workers (*n* = 42) Non-exposed (*n* = 55)	SRQ: Work history	Blood lead concentration Thyroid function-TSH/ FT4/FT3 Thyroid volume-ultrasound	Medium
Baloch et al. Pakistan (2020) [58]	Metals: Lead, Cadmium	Battery recycling, welding	12–18 years	M	12–18 years Battery workers (*n* = 95) Welding workers (*n* = 60) Non-exposed (*n* = 100) 20–45 years Battery workers (*n* = 100) Welding workers (*n* = 120) Non-exposed (*n* = 145)	SRQ: Current work status/work history ENV: Water, soil *****	Blood/hair lead/cadmium Haemoglobin	Medium
Bose-O’Reilly et al. Indonesia/ Zimbabwe (2008) [71]	Metals: Mercury	Gold mining	9–17 years	M/F	Exposed workers (*n* = 80) Children living in exposed areas (*n* = 80) Non-exposed (*n* = 50)	SRQ: Current work status/work history	Blood/urine/hair mercury Medical symptoms Neurobehavioural tests	Medium
Junaid et al. Pakistan (2017) [72]	Metals: Various	Leather and surgical instrument manufacturing	8–18 years	M/F	Exposed workers (*n* = 60) Unexposed (*n* = 15)	SRQ: Current work status/work history ENV: Equation to measure exposure from inhalation, ingestion and dermal contact	Blood/urine/saliva/hair various metals	Medium
Sughis et al. Pakistan (2014) [73]	Metals: Various	Carpet weaving, brick industry	8–12 years	M/F	Carpet weaving workers (*n* = 80) Brick industry workers (*n* = 80) School: high air pollution area (*n* = 100) School: lower air pollution area (*n* = 79)	SRQ: Current work status/work history ENV: Water and particulate matter	Urine various metals	Medium
Kazi et al. Pakistan (2015) [57]	Metals: Lead	Battery recycling	12–15 years	M	Exposed workers (*n* = 118) Exposed non-workers (*n* = 85) Non-exposed (*n* = 90)	SRQ: Work history	Hair lead	Low
Tiwari et al. India (2012) [16]	Metals: Chromium	Gem polishing	<14 years	M/F	Gem polishing workers (*n* = 24) Non-workers (*n* = 23)	SRQ: Current work status/work history	Blood chromium	Low
Lahiry et al. Bangladesh (2011) [14]	Metals: Various	Waste disposal	8–15 years	M/F	Waste disposal workers (*n* = 20) Non-workers (*n* = 15)	SRQ: CL status	Oxidative stress DNA damage-Comet assay Liver function tests	Low
Cuadra et al. Nicaragua (2006) [74] ****	POPs: Various	Waste disposal	11–15 years	M/F	Workers living onsite (*n* = 11) Workers living nearby (*n* = 23) Non-workers living nearby (*n* = 16) Non-workers living nearby/not eating lake fish (*n* = 16) Non-workers living remotely/not eating lake fish (*n* = 11)	SRQ: Current work status/work history/diet	Serum POP POP metabolites	Medium
Athanasiadou et al. Nicaragua (2008) [15] ****	POPs: PBDE	Waste disposal	11–15 years	M/F	Workers living onsite (*n* = 19) Workers living nearby (*n* = 44) Non-workers living nearby (*n* = 31) Non-workers living nearby/not eating lake fish (*n* = 18) Non-workers living remotely/not eating lake fish (*n* = 19)	SRQ: Current work status/work history/diet	Serum PBDE PBDE metabolites	Medium

CL = Child labourer. SRQ: Self-report questionnaire. ENV: Environmental assessment. BIO: Biomarker. * 10-month cohort study. All others are cross-sectional. ** One participant was 19 years old. *** Papers from the same overall study. **** Papers from the same overall study. ***** Authors used environmental assessments as outcomes, rather than as exposures in these two studies.

**Table 5 ijerph-18-05496-t005:** Key areas of future research.

Category	Key Areas
Health outcomes	No studies looked at acute poisoning cases for example, despite a high incidence of work-related poisoning cases in children, both in the developed [120] and developing world [121]. Neurobehavioural deficits due to lead exposure. Long-term health outcomes e.g., chronic conditions in adulthood and diseases with long latencies, such as cancers.
Chemical types	Ammonia and chlorine-based bleaches, for example used by child labourers working in domestic cleaning roles. Further pesticide studies are needed due to the high proportion of child labourers working in agriculture.
Industries	Transport and construction were identified as being particularly hazardous to child labourers [10]. No studies were found looking at chemical exposures in these areas.
Countries	Few or no studies were found from China, Africa, Latin America. An expanded database search is recommended for these areas, looking at non-English language papers.

## Data Availability

Data is contained within the article.

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
