# Peer review of "The Health Impacts of Hazardous Chemical Exposures among Child Labourers in Low- and Middle-Income Countries"

_ijerph, 2021, doi:10.3390/ijerph18105496_

Round 1

Reviewer 1 Report

The manuscript refers to review the health impacts of hazardous chemical exposures among child labourers in low- and middle-income countries. Child labor has social and public health impacts. In some workplaces, children are exposed to toxic agents with important health consequences. The subject reviewed by the authors is important. The manuscript is pleasantly written, but, but some points deserve attention.

1- I suggest that authors do not use acronyms in the title, such as LMIC. Replace LMIC for “low- and middle-income countries”.

2- There are a lot of acronyms in the manuscript. I suggest that the authors make a list or a table of the acronym means.

3- I suggest that the authors define low- and middle-income countries. Perhaps the World Bank classification may be useful.

4- Page 3, “Children are not little adults” section. It is not usual to reference the title of a section. Perhaps the indication of the reference would be better in the text of the section, after the appropriate language adjustments. The term “critical windows of development” should be better explained.

5- Page 22, 4.6 section, last paragraph. “This review may be the first to synthesize….” instead “This review is the first to synthesize…”

Reviewer 2 Report

In the manuscript by Scott et al., the reported evidence in the last two decades on chemical exposure in children laborers and associated adverse health parameters in Low- and Middle-Income Countries were systematically reviewed and summarized, yielding new information. Specifically, abnormal biomarkers and cellular and molecular alterations were identified with certain. Chemical exposure. Overall, this study was well designed with a thorough review and analysis on relevant literature. The methods employed were appropriate and results properly interpreted. The reviewer only has some minor concerns as outlined below.

Minor concerns:

  1. As the first of this study’s objectives to “identify common chemical exposure and adverse health consequences”, in section 4.2, it will be helpful to categorize the biological parameters into relevant health areas, such as cancer/malignancy, immunological/autoimmune diseases, metabolic disorders, etc., which then can be easily correlating to potential adulthood diseases.

  1. The last paragraph of section 3.3.3, was not clear. The mean blood lead measurement in battery recycling child labors and welders were not mentioned. It was not clear how comparison could be made among pottery production, battery recycling and welder.

Reviewer 3 Report

In this manuscript, Scott and Pocock reviewed 23 previously conducted researches regarding chemical exposure and adverse health consequences among children laborer in Low-and Middle-Income Countries (LMIC). The authors screened multiple scientific databases for the review.

The authors provided tables and precisely designed figures for readers and appendixes containing detailed information about the reviewing process.

I, however, would like to make some comments for the authors to consider.

Major comments 

Title and abstract 

  • I suggest the authors go over the journal instruction thoroughly. According to the instruction, “the abstract should be a single paragraph (200 words MAXIMUM) and should follow the style of structured abstract, but WITHOUT headings.” The abstract should be shortened accordingly.
  • I believe using an abbreviation (LMIC) in the title should be avoided.

Introduction 

1.1 Rationale

  • I believe the rationale of this study can be better explained by reconsidering the orders of the sentences. It seems better to provide definition of hazardous child labor in the beginning of the paragraph, followed by the size of the industry. In addition, it seems relevant to me that “Child labor have minimal occupational …… that it is illegal in many settings” should appear prior to “Children face varied occupational risks, …… , workplace violence and low or no wages”

  • Considering the statement “Child labor is most prevalent in Africa and the Asia-Pacific Region, where nearly 50% of child labors are 5-11 years old”, and since the study contains review results of few studies conducted in the South America as well, it would be appropriate to mention some countries of the continent as well as Africa and Asia-Pacific region in this section.

  • In this section, I think you should explain why you focused on LMIC and excluded studies from HIC.

1.1.1 

  • The last sentence of the first paragraph should be reconsidered. Is it really necessary? Or is it better to add “some” in between “Whilst chemicals”. The statement in the next paragraph, “Chemicals are used in numerous industries for different tasks”, sounds pretty similar as well.

  • Overall, the manuscript deals with four types of chemicals: pesticides, heavy metals, solvents, and POPs. The second paragraph explains about these associated with related industries and tasks. I guess it can be clarified by summarizing the contents according to the types of chemicals. My suggestion is to provide another column ‘Types of chemical (pesticides, heavy metals, solvents, and POPs)’ on the left side of the table 1 and provide ‘specific sorts of chemicals that are commonly used in hazardous child laborers’ environments’ in the second column. Then on the third column, types of industry using the chemicals can be provided, followed by specific tasks (conducted by children) related to the industries. The paragraph can be revised by summarizing the table contents. In addition, some types of POPs noted with asterisks in the table can be better explained in the manuscript. Consider providing simple definition of POPs and examples, for readers who are not familiar with the term.

  • I suggest the authors to reconsider the necessity of the third paragraph here. Mentioning about chemical exposure in general environmental setting is necessary in the manuscript, but I don’t think it should appear in the rationale. It might be better to mention it in the discussion part.

1.1.2

  • It seems this part of the manuscript is trying to describe why chemical exposure among child laborers is unique physically (easier absorption …), emotionally (immature), legally (lack of understanding about related laws and regulations), and socially (low-education, illiterate …) or rather is important to examine, especially compared to adults or non-working children.

  • Although the data from high income countries are more accurate, I don’t think it should be mentioned prior to other statements since this manuscript focuses on LMIC.

  • In addition, the authors tended to explain inappropriateness of environmental conditions (quality of workplace environment) in general, rather than implicating vulnerability of child laborers as a cause of the chemical exposure.

1.1.3

  • Again, I believe the authors should focus on the situations of LMIC. In this paragraph, the following information should be given; 1) harmful chemical exposure threshold of adults in LMIC--> not applicable for children 2) other measuring techniques applying in HIC 3) appropriateness of setting chemical exposure threshold (consider homeostasis of human beings and U-shaped dose-response relationships as well)

  • OSH was used before as an abbreviated form and here it is expanded – Occupational Safety and health.

1.2 Objectives 

  • It seems to me the objective number 3 is the actual objective of this study. Identification can be completed by examining 1&2.

Materials and Methods 

2.1 

  • It seems appropriate to include the librarian as an author?.

  • mesh--> MeSH (Medical Subjective Headings)

  • I couldn’t find appendix 4. Did you mean appendix C on page 28? (Indeed, there are two appendixes C on pages 28 and 30)

2.2 

  • There are some redundancies in between the table 2 and the associated texts and within the table 2 (two rows for ‘participants’). In addition, it seems to me some contents in the table 2 can be shortened (e.g., participants are 18 years of age and under --> <18 years, etc.)

  • According to the statement “The inclusion age group was chosen ……. older children as being between 5-12 years of age and adolescents as 12-18 years of age”, the age limitation of the study is irrelevant – should be <12 years. Compare the details of ref #9 and #2. Otherwise, using a term “child and adolescent laborer” is more appropriate.

2.3

  • I think it is not necessary to categorize this part of the manuscript into three subtitles (2.3.1/ 2.3.2/ 2.3.3). Some details are overlapping, e.g. “Many studies use self-report questionnaires, which provide information on current work hours and work history,… (2.3.2.), “Again, the use of self-report questionnaires can be used to measure some health outcomes (2.3.3.)”

  • Chemical exposure: 

       - Self-report questionnaires appear in the manuscript several times. As you had mentioned in the manuscript, many child laborers have reading problems. Some more explanations about self-reported questionnaires are necessary (some studies used trained interviewers? Participants completed survey sheets with help of others? These factors would have influenced the results of the studies)

        - The definition of biomarkers is somewhat confusing. The fundamental elements from a human body containing toxins and metabolites of chemicals (urine, blood, hair, saliva, etc. ) should be differentiated from the types of materials (toxins and metabolites or DNAs) detected from them.

         - It seems to me the authors mentioned a total of three means of chemical exposure measurement, getting information from self-reported questionnaires, checking the status of biological materials (biomarkers or DNAs), and indirectly examining environmental toxin levels.  

  • Health outcomes

         - First of all, I believe it is paramount of importance to define what you meant by “health impact” in this review article since the objective of this study focuses on it.  

          - In addition, the types of health outcome defined in this manuscript seems not systematically organized. It should be better to create types in orders, e.g., from micro to macro or DNA ~ biomarkers (toxin level ~ metabolite level) ~ organ ~ system, and so on. The health outcome types defined in the manuscript now seems overlapping somehow.

            - I suggest you revise the table 3 and 2.3 considering the above comments.

2.4 

  • Where is appendix X (“(please see appendix X for details)”)? (do you mean appendix C on page 30?)

  • As you mentioned in the limitation of the study, the fact that the literature search was completed by only one author in such a short period of time (3 days?) is one of the problems of this study. To me, it’s probably possible if she had put her best efforts. However, the fact that the quality appraisal was completed by the one who completed the literature review might be another problem. I suggest the corresponding author to go over the results of quality appraisal and approve it.

Results 

  • Table 4 should follow section 3.1.

  • It seems it is possible to organize the table 4 more neatly. Since the authors defined the types of chemicals into four different types (pesticides, heavy metals, solvents, and POPs), I believe it should be organized accordingly. The column, sample description, should be subdivided and give an organized information that can be understood at a glance. I guess it is important to show the total number, range of age, and gender of the study sample separately and distinctively for each study. The present table shows some redundant information by showing specific information in the ‘sample description’ column which can also be found in the ‘industry’ column.

  • As I mentioned above, the means of chemical exposure measurements can be categorized into three types: self-reported questionnaires, checking the status of biological materials (biomarkers or DNAs), and indirectly examining environmental toxin levels. The column, ‘author-defined exposure measurement’, can be subdivided. O/X or abbreviations for the above measurement types can be used to summarize the table more neatly.

  • As I mentioned above, while considering the specific definition of health outcome that this study is dealing with, the authors should also think about the definition of the column, ‘health outcomes’, in the table 4. As I mentioned above, by defining and categorizing health outcome more specifically, this column can give more detailed information. It seems to me the column gives some mixed information. Differentiating body elements from the types of materials detected is essential.

  • I believe it should be helpful for some readers to organize researches in order of study quality appraised within the category of the four types of chemicals (pesticides, heavy metals, solvents, and POPs).

  • Descriptions in 3.3.1, 3.3.2, 3.3.3, and 3.3.4 can be shortened by revising the table 4. (I believe 3.3.2 should be heavy metals and 3.3.3 should be solvents)

  • Moreover, specific descriptions regarding pesticides, heavy metals, solvents, and POPs need to be organized in a better way. As I mentioned above, since this study focuses on health impacts, it should be summarized in a way of delivering information regarding health impacts mainly. You should think about a way to explain health impacts of each chemical exposure among child laborers systematically. There are biological impacts (elevation/decrease of chemical concentrations …), symptomatic impacts (decreased pulmonary functions, wheezing, development of neurologic symptoms …. ), and development of specific diseases.

  • 3.3.5 

    The statement “Interestingly, no adverse health impacts were found ….. from any chemical type” may lead to a misunderstanding since it sounds too strong or too confident. I suggest “studies examined relationship between chemical exposure and pulmonary function, thyroid volume, blood pressure, or liver function among child laborers provided no significant associations” and avoid using “interestingly”. In addition, I am confused by the next sentence. Do you mean that it is difficult to draw conclusions from these studies since only single study is completed for each health impact?  

Discussion 

4.1 & 4.2 

  • By revising the results, 4.1 and 4.2 can be organized in a better way.

4.3

  • Since this review article deals with cross-sectional studies mostly, is it possible to discuss impact of chemical exposure in later life in a separate section? There are some points that is helpful for readers in the section though. You can use some sentences in 4.1 and 4.2 as additional information.

4.4 

  • It seems to me this section provides meaningful information regarding measurement of chemical exposure in child laborers. However, it lacks specific descriptions about research results reviewed in the study. In addition, for each sub-section consider my following comments.

  • 4.4.1 

    Mentioned problems regarding self-report questionnaires can be summarized into 1) misreporting and 2) misclassification. Moreover, misreporting can be caused by literacy skills and age. In my opinion, 3) quality of surveys may also affect study results. Some more information regarding self-report questionnaires might be helpful, e.g., number of questions, contents or field of questions, specificity of the survey, etc. The descriptions can be revised considering this. (Why did you mentioned self-esteem as an example of a word that little kids cannot understand?)

  • 4.4.2 

    Mentioned problems regarding measurement of biological materials include 1) hard to differentiate acute/chronic exposure 2) difficult to assess single biological material 3) sample contamination (hair, saliva) 4) individual variability. I think quality of measurement is one of the key issues. Who examined what at where with what kind of machine? Calibration is completed for all studies? In order to examine concentration of a certain biological material, a single and unique technique is used for every study dealing with the material? Another issue that you should think about is that if a biological material was identified at a high-level, is it always bad? Are there any studies which show that chronic low-level exposure is more detrimental than acute high-level exposure? With regard to sample contamination, is there any possibility that urine or blood can be contaminated as well since reviewed researched were conducted in LMIC.

    In addition, since DNA damage and oxidative stress was checked in some studies, it is necessary to discuss about measurement of these as well.

    About individual variability, why did you only mention cholinesterase?

4.5 --> shouldn’t it be 4.4.3?

  • Issues associated with indirect examination of environmental toxin levels include effects of 1) parents (including transfer from mother to fetus) 2) diet and 3) pollution. Providing some information regarding environmental conditions of some countries where researches reviewed in this manuscript were conducted might be helpful as well.

4.6 --> 4.5

  • Regarding the fact that no studies from China is included in this study, give some evidence that child labor is prevalent in China. In addition, if it is prevalent in China, is it legal due to the country’s political characteristics? How about North Korea then?

4.6--> 4.7

  • Is the table not numbered necessary? Can it be described with texts?

4.7-->4.8 & Conclusion

  • It seems these parts give some general information regarding child laborer. Give some specific conclusion and priorities based on the study results, especially focusing on health impacts of chemical exposure.

Minor comments 

  • There are some abbreviations which need to be identified (e.g., ILO, CDC, TCPy, LSHTM, OSHA ..... etc.)

Reviewer 4 Report

Dear authors,

Your manuscript is interesting but I need you to answer some questions:

INTRODUCTION

  • The Introduction is very long and should not be to have subsections.
  • There should be no figures or tables in the introduction. This is supplemental material that is included in other sections.

MATERIALS AND METHODS

  • The eligibility criteria are unclear:
    • Doctoral theses or reviews were included? The authors do not specify it and they are not "gray literature".
    • The authors do not specify between which years the selected studies are included.
  • The information in sections 2.3.2 and the middle of section 2.3.4 do not go under “material and methods”. This information should go in "Introduction".

RESULTS

  • The authors say they have included 23 studies but there are references 53-73 (20references) in this section. The authors should explain this error.

REFERENCES

  • Many bibliographies are obsolete. The bibliographic citations used are more than 5 years old (61.7 %). The authors must update and arrange the bibliography.
  • Some references do not meet the journal guidelines. The authors should review this section.
  • Some references are incomplete or have errors. The authors should review this section.

Reviewer 5 Report

The paper read well and did set out to highligt research gap around the subject. To help improve the paper quality consider: 

i. Section 3.1.  Consider revising the first paragraph to help clarity  on the point considered. There was mention of twenty three papers from 20 studies but  the following statement did not fully  account the 3 missing papers. 

ii.  It was not clear what yardstick was used to adjudge the quality of the papers as high, medium and low this need be made clear. 

iii. Overal, there is further room to proofread the paper to improve the quality. 

Round 2

Reviewer 3 Report

Thank you for the authors for your detailed responses. 

In the 1st round of the review, I did not realize the number of comments that I've made. I would like to appreciate your endeavor at revision. I'd like to apologize for giving you such a time-consuming job as well. 

The following are some minor points that seemed as unnoticed errors.

1. Pg 3 

  •    1.1.2". Children  ~ ---> I believe it should be 1.1.2. "Children ~ 
  •    Although I don't think I am qualified to judge about English, the statement "Measuring chemical exposure levels children also presents challenges" contains a grammar error. It seems to me there is a preposition missing in between 'levels' and 'children'. (If my point regarding English is not correct, please understand kindly) 

2. pg 6  Table 3

  •  There is a redundancy (two rows for DNA damage at the top and bottom) in the bottom section of the table.

3. pg 7 

  • Again, I might be wrong. Could you be so kind and check if a comma is required in the following statement - after 'hair'-; "Biomarkers are often measured using samples of blood, hair urine and saliva."?

4. pg 10 

  • The authors did provide responses regarding my comments on the study results. However, they've decided to keep most of them as the originally submitted form. I do understand most of their responses and agree to their points. However, I suggest the authors to re-think about the way of presenting table 4.
  • Although the authors tried to make amendments by presenting researches in the order of study appraised within the category of the four types of chemicals (pesticides, solvents, heavy metals, and POPs) as my suggestion, I would like the authors to consider about my other comments regarding Table 4 once again. I still believe the contents of the table can be presented in a better way.  If  you were to keep the horizontal alignment of the document form, space constraints cannot be a problem, I think. 

5. Pg 23 

  • At the end of the first paragraph, I believe, it is necessary to check the citation form. Shouldn't it be [29],[116]?  Moreover, in the author's response document (page 3), you've cited [117],[118]. Please check the appropriateness of the citation use.  

6.  Pg28 Appendix B1

  •   mesh --> MeSH

7. pg 29 Appendix B2 

  • Medline --> MEDLINE (please check the footnote as well)

8. pg 34 The list of acronyms 

  •    I've noticed some abbreviations that were not included in the list, e.g., CDC, IARC, UNICEF, etc. I guess you've missed these since these are familiar terms. 

The followings are some points I want you to consider a little bit. 

  • regarding the exclusion of studies from HICs, I think it is better to mention about the recent reviews previously reported only, except for the time constraint. 
  • I think appendix A~ B2 would be easier to read if these part of the document are presented in the vertical alignment. 
  • I think the authors should decide what to do about presenting the orders of the four types of chemicals. The orders are inconsistent throughout the manuscript now, e.g., in table 1, pesticides -heavy metals-solvents-POPs  versus ; in the result, 3.3.1 pesticides -3.3.2 solvents -3.3.3 heavy metals -3.3.4 POPs, etc. ). Check the orders throughout the manuscript (text, table and figures) for uniformity.  

Reviewer 4 Report

Dear authors,

Except for minor changes you have ignored the indications. Also, they have not given solid arguments to refute my indications.

MATERIALS AND METHODS

  • The doctoral theses are original research work. By definition, they are a primary source. In the following link you can see how a university specifies it:

https://libguides.umn.edu/c.php?g=986651

RESULTS

  • The number of studies is irrelevant. If they are different publications, they must be numbered. It is not logical that you say that there are 23 and there are only 20 results.

REFERENCES

  • You are right, I was wrong. Checking the figures excluding the references from the "results" the manuscript has many outdated references. The bibliographic citations used are more than 5 years old (64.8 %). The "results" cannot be chosen but the updated references to elaborate the "introduction" and the "discussion" if you can choose them.

Seeing this, I am sorry but I have no choice but to reject your manuscript.

Best regards

Reviewer 5 Report

Dear Authors, 

Manythanks for taking time out to respond to my observation made to the paper. 

I am happy to accept the corrections made.  

Best regards 

Reviewer
